# Friends in Unexpected Places: Enhancing Local Fairness in Federated Learning through Clustering

## Abstract

Federated Learning (FL) has been a pivotal paradigm for collaborative training of machine learning models across distributed datasets. In heterogeneous settings, it has been observed that a single shared FL model can lead to low local accuracy, motivating *personalized FL* algorithms. In parallel, fair FL algorithms have been proposed to enforce group fairness on the global models. Again, in heterogeneous settings, global and local fairness do not necessarily align, motivating the recent literature on *locally fair FL*. In this paper, we propose new FL algorithms for heterogeneous settings, *spanning the space between personalized and locally fair FL*. Building on existing clustering-based personalized FL methods, we incorporate a new fairness metric into cluster assignment, enabling a tunable balance between local accuracy and fairness. Our methods match or exceed the performance of existing locally fair FL approaches, without explicit fairness intervention. To support this finding, we demonstrate (numerically and analytically) that personalization *alone* can improve local fairness and argue that our methods exploit this alignment when present.

## 1 Introduction

Federated Learning (FL) has been the pivotal paradigm for collaboratively training machine learning models across distributed datasets/clients in a privacy-preserving manner (Kairouz et al., 2021). Shared, *global* models learned through FL can effectively aggregate gradient information from multiple clients, and (potentially) outperform *standalone* models trained individually by each client in the absence of collaboration. However, when clients have heterogeneous datasets, the convergence speed of FL algorithms can considerably deteriorate (Li et al., 2019b; Zhao et al., 2018), and clients with less typical data distributions may experience low local accuracy (Tan et al., 2022; Karimireddy et al., 2020; Li et al., 2020). To address these limitations, a spectrum of *personalized FL* techniques have been proposed (e.g., (Li et al., 2020; Ghosh et al., 2020; Briggs et al., 2020; Sattler et al., 2020; Fallah et al., 2020; Mansour et al., 2020)). These enhance the *local accuracy* of the learned models while keeping some of the benefits of collaborative learning.

Beyond ensuring model accuracy, it has become increasingly important to train machine learning models that satisfy *(group) fairness* (Barocas et al., 2019). This means ensuring that the learned models treat individuals from different demographic groups equally by, e.g., maintaining similar selection rates or true positive rates across groups defined by sensitive attributes (e.g., race, gender). Motivated by this, a number of works have proposed *fair FL* algorithms; see Shi et al. (2023); Salazar et al. (2024) for surveys. However, most of these existing works focus on *global* fairness: fairness of a shared, global model assessed on the global data distribution. Such model is not necessarily *locally* fair when clients have heterogeneous datasets. For example, both population demographics and healthcare data distributions vary geographically (Swift, 2002). Given this heterogeneity, a fair federated learning model trained across state hospitals can satisfy a desired fairness criterion at the state level, but may still be unfair at individual hospitals if local demographics differ significantly from the overall population. To address this challenge, recent works have formally studied tradeoffs between local and global fairness (Hamman & Dutta, 2023), and proposed FL algorithms that can achieve local fairness (Meerza et al., 2024; Zhang et al., 2025; Makhija et al., 2024; Zhou & Goel, 2025).

In this paper, we are similarly interested in federated learning algorithms that can perform well *locally* when clients have heterogeneous datasets. In particular, *we propose federated learning algorithms that span the range between personalized federated learning and locally fair federated learning algorithms.* See Figure 1 for an illustration. In this plot, the $x$-axis and $y$-axis show the local fairness gap and the local accuracy, respectively. The plot highlights that personalized FL algorithms (Ghosh et al., 2020; Briggs et al., 2020; Sattler et al., 2020; Fallah et al., 2020; Li et al., 2020) improve local accuracy at the expense of fairness, whereas locally fair FL algorithms (Makhija et al., 2024; Zhou & Goel, 2025) sacrifice local accuracy to improve fairness guarantees locally.

To span this gap between personalized and locally fair FL algorithms, our proposed algorithms, `Fair-FCA` and `Fair-FL+HC`, take inspiration from clustering methods used for personalization in FL (Ghosh et al., 2020; Briggs et al., 2020).

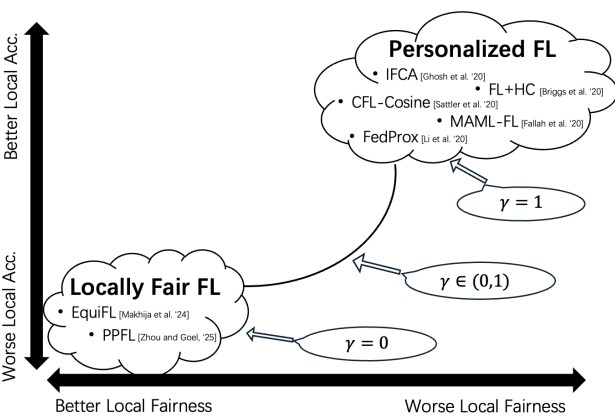

Figure 1: Our proposed algorithms span the gap between personalized FL and locally fair FL methods, and lead to a new class of locally fair methods. The parameter $\gamma \in [0, 1]$ adjusts the balance between local accuracy and local fairness.

They integrate a new fairness metric into the cluster assignment step, allowing us to improve local fairness while balancing accuracy and fairness at the client level. Our methods can be viewed as a way of strategically grouping either "similar" or "useful" clients together, as appropriate, given the desired local fairness-accuracy tradeoff. Importantly, depending on the desired local fairness-accuracy tradeoff, these may end up being the clients with the most similar or the most complementary (in terms of demographic statistics) data distributions.

Specifically, our algorithms use a parameter $\gamma \in [0, 1]$ to adjust the balance between local accuracy and local fairness during cluster assignment, with larger $\gamma$'s indicating more focus on local accuracy. At $\gamma = 1$, our proposed algorithms, `Fair-FCA` and `Fair-FL+HC`, reduce to the existing `IFCA` (Ghosh et al., 2020) and `FL+HC` (Briggs et al., 2020) algorithms which they are built on. At $\gamma = 0$, our algorithms are of independent interest, as they present two new algorithms for improving local fairness in FL. In particular, existing approaches to locally fair FL (Meerza et al., 2024; Zhang et al., 2025; Makhija et al., 2024; Zhou & Goel, 2025) achieve their goals through *explicit* fairness interventions during or after training: they either prevent bias propagation during collaborative training by mitigating the local unfairness or reducing the influence of biased clients via weighted aggregation, or modify predictions probabilistically (post-training) to satisfy fairness constraints. In contrast, we show that our methods can achieve comparable and at times better local fairness by simply clustering clients *without any additional interventions* (i.e., without pre-, in-, or post-processing steps), while incurring comparable (or slightly lower) runtime. We also note that our proposed clustering-based approach to achieving local fairness is *orthogonal*/complementary to explicit fairness interventions; that is, our proposed methods can be used in conjunction with existing approaches to locally fair FL.

Finally, we explore the connections between personalization and local fairness in FL to provide intuition for the efficacy of our proposed approach. Specifically, we show that personalization *alone* can enhance local fairness. We illustrate this alignment between personalization and fairness through extensive numerical experiments on a range of personalized FL methods (clustering (Ghosh et al., 2020; Briggs et al., 2020; Sattler et al., 2020), local-finetuning (Fallah et al., 2020), and regularization-based (Li et al., 2020)), on real-world (`Adult` (Dua & Graff, 2017) and `Retiring Adult` (Ding et al., 2021)) and synthetic datasets, and for several notions of group fairness (statistical parity, equality of opportunity, and equalized odds (Barocas et al., 2019)). We highlight two potential factors driving the alignment: *statistical* advantages of increased data diversity due to the collaborative nature of FL, and *computational* advantages when there is an alignment in local accuracy and fairness. We note that our proposed algorithms are effectively taking advantage of the alignment of personalization and fairness, whenever possible, to improve the local fairness-accuracy tradeoff.

**Summary of findings and contributions.**

*1. New fairness-aware and personalized federated learning algorithms.* We propose two new algorithms, `Fair-FCA` and `Fair-FL+HC`, which take both local accuracy and local fairness into account when (iteratively) determining clients' cluster memberships, allowing for a tunable fairness-accuracy trade-off at the client level.

*2. New clustering-based algorithms for improving local fairness.* Our proposed tuneable algorithms lead to new methods for locally fair FL. In contrast to existing methods for locally fair FL, our methods do not include any explicit fairness interventions (pre-, in-, or post-processing steps) yet can achieve comparable or better local fairness by simply clustering clients. We highlight how our methods are orthogonal to existing approaches to local fairness in FL, and can therefore be used simultaneously to further enhance local fairness guarantees.

*3. Unintended fairness benefits of personalization.* We conduct extensive numerical experiments, under different notions of fairness, and using both real-world and synthetic data, to show that personalization *alone* can improve local fairness as an unintended benefit. This is an alignment our algorithms are exploiting when possible. We highlight the potential statistical and computational reasons leading to this alignment, and provide analytical support under certain conditions (Propositions 1 and 2).

## 2 Related Work

Our work is at the intersection of two lines of literature: personalized FL and fairness in FL. We review works on personalized FL including clustering-based methods (and other related work) in Appendix B. In terms of fairness in FL, we note that this term has taken different interpretations in the FL literature. Much of the early works in fair FL (Li et al., 2019a; 2021b; Zhang et al., 2021; Wang et al., 2021) primarily focused on *performance fairness*, which seeks to achieve uniform accuracy across all clients. However, even if a trained model attains uniform performance, it can still exhibit bias against certain demographic groups. Motivated by this, we focus on notions of *group fairness* in FL (Barocas et al., 2019; Salazar et al., 2024) which aim to attain similar performance of the learned models across protected groups. Even within this literature, the majority of the works have focused on improving *global* group fairness (Abay et al., 2020; Gálvez et al., 2021; Wang et al., 2023; Zeng et al., 2021; Ezzeldin et al., 2023; Liu et al., 2025). In contrast, we focus on *local* group fairness.

A number of recent works have also explored algorithms for improving local group fairness in FL (Meerza et al., 2024; Zhang et al., 2025; Makhija et al., 2024; Zhou & Goel, 2025). Meerza et al. (2024) integrate local fairness constraints with fairness-aware clustering-based aggregation, leveraging Gini coefficients to jointly enhance both global and local group fairness. Similarly, Zhang et al. (2025) incorporate locally fair training using the EM algorithm and adjust aggregation weights through reweighting based on distance to achieve fair aggregation. Makhija et al. (2024) enforce fairness constraints in the local optimization problem to prevent bias propagation during collaboration. Zhou & Goel (2025) introduce fairness post-processing techniques (model output fairness post-processing and final layer fairness fine-tuning) to improve local fairness. Compared to these works, our approach to attaining local fairness is different: we demonstrate that improved local fairness, along with a better fairness-accuracy tradeoff, can be achieved by clustering clients based on a fairness-aware assignment metric (or even through personalization alone).

Lastly, some existing works have, similar to ours, noticed connections between fairness and personalization/clustered FL techniques, but they differ from ours in either their notion of fairness, context, or scope. Wang et al. (2024) use healthcare data to demonstrate that Ditto (a personalized FL algorithm by Li et al. (2021b) which enhances performance fairness) achieves better local group fairness compared to standalone learning. Nafea et al. (2022) add a notion of fairness into cluster identity assignment (similar to us) but to ensure proportional fairness among protected groups (a notion different from group fairness). Kyllo & Mashhadi (2023) examine the impact of fairness-unaware clustering on a number of fairness notions. In contrast, we study how and why a *range* of personalized FL algorithms (not just clustering-based) may improve local fairness.

# 3 Fairness-Aware Federated Clustering Algorithms

Our goal is to develop FL algorithms that strike a tunable balance between local accuracy and local fairness, spanning the range between personalized FL and locally fair FL methods (while also presenting a new approach to locally fair FL). To this end, we start from clustering-based personalized FL approaches, which improve local accuracy in heterogeneous settings. The underlying idea of these methods is that clients with similar data distributions (assessed based similarity of their model performance (Ghosh et al., 2020), model parameter (Briggs et al., 2020), or gradient updates (Sattler et al., 2020)) will benefit from forming smaller "teams" together. Inspired by this, *we instead allow clients to join forces based on both fairness and accuracy benefits*, which as we show, may be due to them having similar *or* (appropriately) different local datasets.

We illustrate the viability of this idea by proposing `Fair-FCA` and `Fair-FL+HC`, which build on the existing `IFCA` algorithm (Ghosh et al., 2020) and `FL+HC` algorithm (Briggs et al., 2020), respectively. We choose these algorithms as starting points, since many existing algorithms in the clustered FL literature are built on the `IFCA` framework (e.g., Li et al. (2021a); Chung et al. (2022); Huang et al. (2023); Ma et al. (2024)) and the `FL+HC` framework (e.g., Jothimurugesan et al. (2023); Luo et al. (2023); Li et al. (2023); Sun et al. (2024)).

## 3.1 Problem setting and preliminaries

We consider an FL setting with $N$ clients, where each client $i$ is tasked with a binary classification problem. The client's dataset consists of samples $z = (x, y, g)$, where $x \in \mathbb{R}^d$ represents the feature vector, $y \in \{0, 1\}$ is the true label, and $g \in \{a, b\}$ is a binary protected attribute (e.g., race, sex). A client $i$ has access to $n_i$ such samples, $Z_i := \{z_{ij}\}_{j=1}^{n_i}$, drawn independently from a joint feature-label-group distribution with probability density functions $\mathcal{G}_g^{y,i}(x)$. These distributions differ across clients, which causes the conflicts between model accuracy/fairness at local and global levels.

*Evaluating local accuracy.* Let $f(z, \theta)$ denote the loss function associated with data point $z$ under model $\theta$. This could be, for instance, the misclassification loss. Then, the local empirical loss of client $i$ is given by $F(Z_i, \theta) := \frac{1}{n_i} \sum_{j=1}^{n_i} f(z_{ij}, \theta)$.

*Evaluating local fairness.* Consider a learned model $\theta$, and let $\hat{y}(\theta)$ denote the labels assigned by it. We assess the *group fairness* of $\theta$ according to three commonly studied notions of group fairness.

1. *Statistical Parity* (`SP`) (Dwork et al., 2012) assesses the gap between the selection rate of each group: (i.e., $\Delta_{\texttt{SP}}(\theta) := |\mathbb{P}(\hat{y}(\theta) = 1 | g = a) - \mathbb{P}(\hat{y}(\theta) = 1 | g = b)|$;

2. *Equality of Opportunity* (`EqOp`) (Hardt et al., 2016) finds the gap between true positive rates on each group: (i.e., $\Delta_{\texttt{EqOp}}(\theta) := |\mathbb{P}(\hat{y}(\theta) = 1 | g = a, y = 1) - \mathbb{P}(\hat{y}(\theta) = 1 | g = b, y = 1)|$);

3. *Equalized Odd* (`EO`) (Hardt et al., 2016) is set to the gap between true positive or false positive rates between groups, whichever larger (i.e., $\Delta_{\texttt{EO}}(\theta) := \max_{i \in \{0,1\}} |\mathbb{P}(\hat{y}(\theta) = 1 | g = a, y = i) - \mathbb{P}(\hat{y}(\theta) = 1 | g = b, y = i)|$).

Here, the probability is with respect to the data distributions $\mathcal{G}_g^{y,i}(x)$ of client $i$. These fairness metrics can be evaluated empirically on the client's data realization $Z_i$ (setting the probabilities to the number of data points satisfying its condition divided by the group sample size). Let $\Psi^f(Z_i, \theta)$ denote the empirical local fairness of model $\theta$ for fairness metric $f \in \{\texttt{SP}, \texttt{EqOp}, \texttt{EO}\}$, assessed on $Z_i$.

## 3.2 Integrating fairness metrics in cluster identity assignment

**The `Fair-FCA` algorithm.** Our first proposed algorithm iterates over two steps: (1) cluster identity assignment, and (2) training of cluster-specific models. Specifically, let $\Theta_k^t$ denote cluster $k$'s model at time step $t$. The cluster identity for client $i$ at time $t$, denoted $c^t(i)$, is determined by:

$$c^t(i) = \arg\min_{k \in [K]} \gamma F(Z_i, \Theta_k^t) + (1 - \gamma)\Psi^f(Z_i, \Theta_k^t) \tag{1}$$

---

**Algorithm 1:** `Fair-FCA`

---

**Input**: Number of clusters $K$, number of clients $N$, number of local updates $E$, cluster model
initialization $\Theta_{1:K}$, learning rate $\eta$, fairness-accuracy tradeoff $\gamma$, fairness $f \in \{\mathtt{SP}, \mathtt{EqOp}, \mathtt{EO}\}$.

**Initialize**: Start clusters $k \in [K]$ by randomly selecting one client for each.

**while** *not converge* **do**

    **for** *client $i \in [n]$* **do**

        **Find** cluster identity:

            $c(i) = \arg\min_{k \in [K]} \gamma F(Z_i, \Theta_k) + (1 - \gamma)\Psi^f(Z_i, \Theta_k)$

        **Initialize** $\theta_i = \Theta_{c(i)}$

        **Perform** $E$ steps of local update

            $\theta_i = \theta_i - \eta\nabla_{\theta_i} F(Z_i, \theta_i)$

        **Upload** $\theta_i$ to server

    **end**

    **Update** the cluster model $\Theta_{1:K}$

        $\Theta_k = \Theta_k - \sum_{i \in C_k} \frac{n_i}{\sum_i n_i}(\Theta_k - \theta_i)$

    **Send** new cluster models $\Theta_{1:K}$ to all clients

**end**

**Output**: Cluster models $\Theta_{1:K}$, Cluster identity $c(i), \forall i \in [n]$.

---

Here, $K$ be the total number of clusters (a hyperparameter), and $\gamma$ is a hyperparameter that strikes a desired balance between accuracy and fairness. For $\gamma = 1$, we recover the `IFCA` algorithm; for $\gamma = 0$, we obtain a clustered FL algorithm that prioritizes only (local) $f$-fairness when grouping clients. For $0 < \gamma < 1$, we obtain clusters that provide each client with the best fairness-accuracy tradeoff among those attainable if the client were to join each cluster.

Let $C_k^t$ be the set of clients whose cluster identity is $k$ at the end of this assignment process (i.e., $C_k^t = \{i \in [n] : c^t(i) = k\}$). Once clients get assigned clusters, each client $i$ starts from its corresponding cluster model $\Theta_{c^t(i)}^t$, and locally runs gradient steps $\theta_i^t = \Theta_{c^t(i)}^t - \eta\nabla_{\theta_i} F(Z_i, \Theta_{c^t(i)}^t)$ to update it. Then, the updated local models $\theta_i^t$ are sent to the central server, who uses these to update the cluster models to $\Theta_{1:K}^{t+1}$ by taking the weighted average of the local models of clients in corresponding clusters. Formally, $\Theta_k^{t+1} = \Theta_k^t - \sum_{i \in C_k^t} \frac{n_i}{\sum_i n_i}(\Theta_k^t - \theta_i^t), \forall k \in [K]$. The pseudo-code for `Fair-FCA` is shown in Algorithm 1.

**The `Fair-FL+HC` algorithm.** In our second proposed algorithm, the algorithm first runs the regular `FedAvg` procedure for a predetermined number of rounds before clustering. Once a global model $\theta^{FA}$ is obtained, each client receives the model and performs several local updates to personalize their local models $\theta_i$.

Like the `FL+HC` algorithm, the `Fair-FL+HC` also employs hierarchical clustering with a set of hyperparameters $P$ to group clients by minimizing intra-cluster variance, measured using the $L_2$ Euclidean distance metric, as well as fairness performance.

$$\text{Clusters} = \text{HierarchicalClustering}(\gamma\mathbf{D} + (1 - \gamma)\mathbf{\Psi}^f(Z, \theta), P) \tag{2}$$

Here, $\mathbf{D}$ is a symmetric matrix where each entry $D_{i,j}$ represents the Euclidean distance between $\theta_i$ and $\theta_j$. $\mathbf{\Psi}^f(Z, \theta) := \max(\Psi^f(Z_i, \theta_j), \Psi^f(Z_j, \theta_i))$, with $f \in \{\mathtt{SP}, \mathtt{EqOp}, \mathtt{EO}\}$ is also a symmetric matrix that captures the worst-case $f$-fairness performance when client $i$'s model is evaluated on client $j$'s local data, or vice versa. The parameter $\gamma$ balances between fairness and accuracy considerations; when $\gamma = 1$, we recover the `FL+HC` algorithm. Once clustering is completed, each cluster trains its model independently using `FedAvg`. The pseudo-code for `Fair-FL+HC` is shown in Appendix C.

### 3.3 Comparison with existing locally fair FL algorithms ($\gamma = 0$)

We first benchmark our proposed methods against the baseline `FedAvg` algorithm (McMahan et al., 2017), as well as two fair FL algorithms specifically designed to improve local group fairness: `EquiFL` (Makhija et al., 2024) and `PPFL` (Zhou & Goel, 2025). The `EquiFL` algorithm (Makhija et al., 2024), an in-processing

approach, enforces fairness constraints during local model training. The PPFL algorithm (Zhou & Goel, 2025), a post-processing approach, probabilistically adjusts model predictions after training to satisfy fairness criteria.

We begin our experiments with a synthetic dataset, focusing on average local statistical parity (SP) fairness. We randomly generate 6 clients, each with different levels of imbalance in the number of samples (e.g., balanced, mildly imbalanced, highly imbalanced). Since SP fairness depends on both group/label rates and data features, the synthetic setting allows us to isolate and analyze the impact of each factor on fairness performance one at a time. Additional experiments using different parameter settings and real-world datasets, and full experimental details are provided in Appendix E.2.

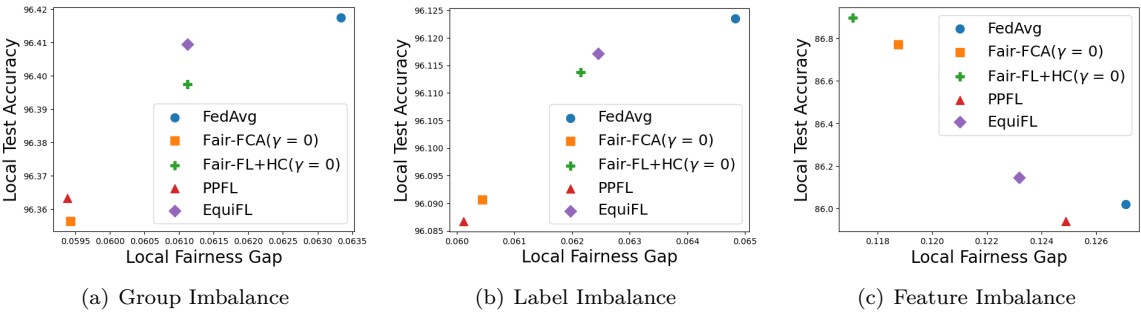

Figure 2: `Fair-FCA` and `Fair-FL+HC` on synthetic datasets ($\gamma = 0, f = $ SP)

Our results in Fig. 2 show that clustering clients by their local fairness metrics improves the fairness performance compared with `FedAvg` and achieves performance comparable to that of existing methods. Interestingly, when group or label rates are imbalanced, `Fair-FCA` outperforms `Fair-FL+HC` in terms of local fairness. We attribute this to bias propagation during the warm-start phase in the `Fair-FL+HC`, which tends to group highly imbalanced clients with mildly imbalanced ones. In contrast, when feature distributions are imbalanced, `Fair-FL+HC` yields better fairness performance than `Fair-FCA`. We believe this is due to its hierarchical clustering approach, which is more sensitive to relative differences in client performance and thus better captures distributional mismatches.

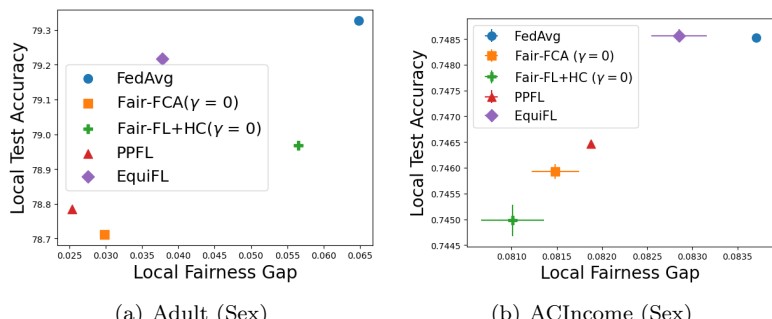

Figure 3: `Fair-FCA` and `Fair-FL+HC` on *Adult* and *Retiring Adult* datasets ($\gamma = 0, f = $ SP)

This insight is further validated by experiments on the `Adult` (Dua & Graff, 2017) and `Retiring Adult` (Ding et al., 2021) datasets, shown in Fig. 3. The `Adult` dataset experiments aim to predict whether an individual earns more than $50k annually based on demographic and socioeconomic features. We randomly generate 5 clients, each with different levels of imbalance in the number of samples based on sex. To explore the impact of more complex distributional differences, we also evaluate on the `Retiring Adult` dataset, which includes census data from all 50 U.S. states and Puerto Rico. Each state is treated as a client, with data samples consisting of multi-dimensional feature vector $x$ (e.g., age, education, citizenship), a true label $y$, and a protected attribute $g$ (e.g., sex). To amplify distributional differences, we manually scale the feature set ($x$) by 60% for the states with IDs $\{1, 10, 20, 30, 40, 50\}$. Our experiments here show the average SP fairness

on the ACSIncome (Income) classification task. We again observe that our proposed methods achieves local fairness comparable to, or better than, that of existing locally fair FL methods.

### 3.3.1 Runtime comparison

Our findings so far establish that our proposed methods achieve local fairness comparable to, or at times better than, that of existing locally fair FL methods. As noted in the introduction, our methods' different approach to achieving local fairness can also enable this performance with a lower computational cost. Intuitively, this is because our methods (e.g., `Fair-FCA`($\gamma = 0$)) do have to compute fairness violations to determine cluster memberships (increasing runtime), but afterwards, each client only needs to do *unconstrained* gradient updates during training. This is in contrast to the per-update cost of satisfying fairness constraints during training when using explicit interventions like in-processing methods (e.g., that of `EquiFL`).

Table 1: Runtime performance comparison. Unit: second.

| `FedAvg` (McMahan et al., 2017) | `EquiFL` (Makhija et al., 2024) | `Fair-FCA`($\gamma = 0$) |
|---|---|---|
| 1594.63 (+/-51.24) | 1890.84 (+/-70.92) | 1806.54 (+/-68.37) |

To support this, Table 1 compares the runtime of our proposed algorithm against the `FedAvg` and `EquiFL` algorithms. All experiments are conducted in the Jupyter Notebook 6.4.12 with Python 3.9.13 on MacBook Pro with 1.4 GHz Quad-Core Intel Core i5 processor and Intel Iris Plus Graphics 645 1536 MB. As shown in Table 1, the `FedAvg` algorithm achieves the fastest runtime, as expected, as it has no fairness consideration. Both our `Fair-FCA`($\gamma = 0$) performing fairness-aware clustering and the `EquiFL` method enforcing local fairness constraints take longer time compared to the `FedAvg` algorithm. Nonetheless, we observe that our algorithm requires slightly less time (4.5%) compared to the `EquiFL` algorithm.

### 3.3.2 Orthogonality of fairness-aware clustering and explicit fairness intervention

We view one of this paper's main contributions as demonstrating that fairness-aware clustering represents a distinct and complementary mechanism to traditional approaches for achieving fairness in FL (whether globally or locally) through explicit fairness constraints during training, while maintaining comparable or slightly improved local fairness and runtime. That said, these two mechanisms are *orthogonal*, meaning that employing one does not preclude the use of the other. To illustrate the feasibility of this idea, we evaluate our fairness-aware clustering algorithm when combined with in-processing fairness methods (similar to the `EquiFL` approach). We observe that in the ACSIncome task with sex as the protected attribute, the combination of fairness-aware clustering and in-processing fairness does not necessarily outperform either `EquiFL` or `Fair-FCA` ($\gamma = 0$) in terms of local fairness, though it

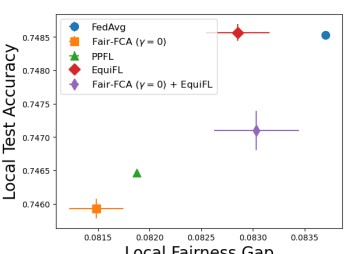

Figure 4: Combining fairness-aware clustering and in-processing fairness interventions.

does attain a different point on the fairness-accuracy tradeoff curve. Further investigation into whether and how to combine the different approaches could be an interesting direction of future work.

### 3.4 Tuneable fairness-accuracy tradeoff using `Fair-FCA` and `Fair-FL+HC` with $\gamma \in (0, 1)$

In addition to presenting a new class of locally fair FL algorithms, our proposed approach can offer a tunable trade-off between accuracy and fairness at the client level (which, to our knowledge, is not considered or achievable by methods proposed in prior work). We begin by conducting a numerical experiment on a synthetic dataset to illustrate the ability of `Fair-FCA` and `Fair-FL+HC` to strike the desired balance between fairness and accuracy. Additional experiments conducted on real-world datasets are shown in Appendix E.3.

We consider a total of 8 clients that could (potentially) be clustered into two clusters. Among these, 6 clients (Client ID: 2,4,5,6,7,8) have similar data distributions, with 4 clients (Client ID: 4,6,7,8) sharing identical distributions across the two protected groups $a, b$ (low fairness gap). The remaining 2 clients (Client ID: 1,3) have different data distributions compared to the first six, but they also share identical distributions across

the two protected groups. We consider $f = \texttt{SP}$. Let $\gamma_1, \gamma_2 \in [0,1]$ be the hyperparameters of the `Fair-FCA` and `Fair-FL+HC` algorithms, respectively.

Figure 5 shows our proposed algorithms span the gap between personalized FL methods (e.g., IFCA, FL+HC, etc.) and locally fair FL methods (e.g., PPFL, EquiFL). When $\gamma_1 = \gamma_2 = 1$, both `Fair-FCA` and `Fair-FL+HC` prioritize accuracy, recovering existing IFCA and FL+HC methods; by design, this is attained by grouping the 6 clients having similar data distributions together ({1,3} and {2,4,5,6,7,8}). In contrast, when $\gamma_1 = \gamma_2 = 0$, both `Fair-FCA` and `Fair-FL+HC` focus only on `SP` fairness by clustering clients that have identical distributions on the two protected groups together ({2,5} and {1,3,4,6,7,8}), achieving comparable or better local fairness to existing locally fair FL methods. Lastly, by setting $\gamma_1, \gamma_2 \in (0,1)$, we can effectively account for both accuracy and fairness when clustering, covering the gap as desired.

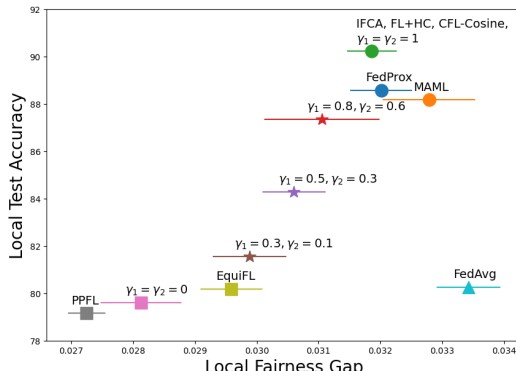

Figure 5: Comparison of our methods with existing personalized FL and locally fair FL methods, under different $\gamma_1, \gamma_2$.

## 4 Personalization alone can also improve fairness

Our findings in Section 3.3 demonstrated that our proposed algorithms effectively improve local fairness by setting their tuneable parameter to $\gamma = 0$. In Section 3.4, we further show that these algorithms attain a tuneable fairness-accuracy trade-off to $\gamma \in (0,1)$. In this section, we now move to the extreme of $\gamma = 1$, which leads to the existing IFCA (Ghosh et al., 2020) and FL+HC (Briggs et al., 2020) algorithms that our methods build on. Interestingly, we show that even without fairness considerations, *personalization alone can still enhance local fairness as an unintended benefit.* We argue that one of the advantages of our proposed methods is that they are exploiting this alignment when present.

We consider several classes of personalized FL methods to illustrate this alignment; this is to show that our insights on the alignment of personalization and fairness hold irrespective of how personalization is achieved. In more detail, we run experiments on the following personalized FL methods. The `IFCA` algorithm (Ghosh et al., 2020) alternates between clustering clients based on model performance and optimizing parameters within each cluster. The `FL+HC` algorithm (Briggs et al., 2020) employs hierarchical clustering to minimize intra-cluster variance, measured by the Euclidean distance between models. The `CFL-Cosine` algorithm (Sattler et al., 2020) partitions clients into two clusters by minimizing the maximum cosine similarity of their gradient updates. Beyond clustering, the `MAML-FL` algorithm (Fallah et al., 2020) extends `FedAvg` by allowing clients to fine-tune the global model through extra local gradient steps. Similarly, the `FedProx` algorithm (Li et al., 2020) adds $l_2$ regularization to balance local and global model learning. We compare the local accuracy and local fairness of these algorithms against FedAvg and standalone learning. We will run these experiments on the `Adult` (Dua & Graff, 2017) and `Retiring Adult` (Ding et al., 2021) datasets, comparing the average local statistical parity (`SP`) fairness achieved by different FL algorithms. Similar experiments supporting our findings under other notions of fairness (`EqOp`, `EO`) are given in Appendix E.4. We then substantiate our findings with analytical support in Section 4.4.

### 4.1 Imbalanced groups: statistical advantages of collaboration

We first consider the ACSEmployment task in the `Retiring Adult` dataset with "race" as the protected attribute. Fig 6(a) shows the fraction of samples in each group-label, from several states, highlighting an imbalance between samples from the White and Non-White groups. This is further evident in Figure 6(b), which shows that most states have only $\sim 10\%$ qualified (label 1) samples from the Non-White group, in contrast to $\sim 35\%$ qualified samples from the White group.

Fig 6(c) shows that all collaborative training algorithms (`FedAvg`, `MAML-FL`, `IFCA`, `FedProx`, `FL+HC`, and `CFL-Cosine`) achieve better local fairness (smaller gap) compared to `Standalone` learning. This is due to the *statistical benefits* of collaboration: each client has limited samples in the non-White group, leading to poorly

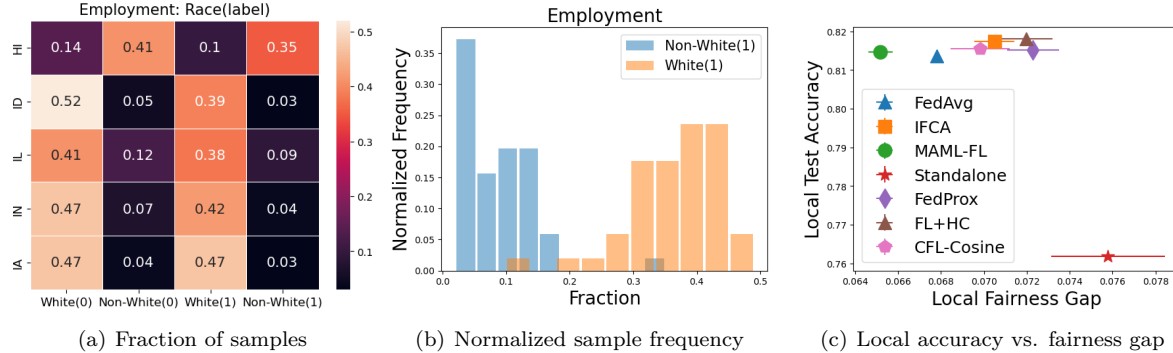

(a) Fraction of samples     (b) Normalized sample frequency     (c) Local accuracy vs. fairness gap

Figure 6: Experiments on the ACSEmployment task with imbalanced groups (race).

trained models with high local fairness gap (and low accuracy). In contrast, collaborative training in essence has access to more data, improving both metrics. For the same reason, the `IFCA`, `FL+HC` and `CFL-Cosine` algorithms, which partition clients into multiple clusters, has (slightly) worse local fairness compared to `FedAvg`. Similarly, the `FedProx` algorithm imposes a regularization term that prevents the local updates from deviating too much from the global model, making it less fair compared to `FedAvg`. In comparison, the `MAML-FL` algorithm, which effectively sees the global dataset (when training the global model that is later fine-tuned by each client), has better local fairness compared to `FedAvg`, indicating that personalization can improve both local accuracy (as intended) and local fairness (as a side benefit).

## 4.2 Better-balanced groups: computational advantages of collaboration

We next consider better-balanced data, to show advantages of collaborative and personalized training beyond the statistical benefits of (effectively) expanding training data. We again consider the ACSEmployment task, but now with "sex" as the protected attribute. Fig 7(a) shows that data samples are more evenly distributed across groups and labels in this problem. Figure 7(b) further confirms that clients exhibit similar sample fractions of label 1 individuals in male and female groups.

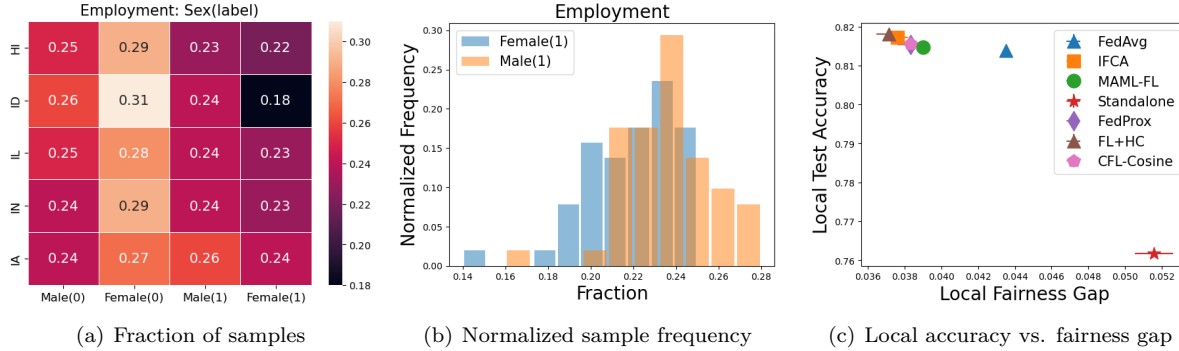

(a) Fraction of samples     (b) Normalized sample frequency     (c) Local accuracy vs. fairness gap

Figure 7: Experiments on the ACSEmployment task with better-balanced groups (sex).

Fig 7(c) shows that all collaborative training algorithms still have better local fairness compared to `Standalone` learning. Furthermore, we observe that all personalized learning algorithms (`IFCA`, `FL+HC`, `CFL-Cosine`, `MAML-FL`, and `FedProx`) improve both local accuracy and local fairness compared to `FedAvg`. This is due to the *computational advantages* of (personalized) collaborative learning: for each client, due to similarity of the data for the male and female groups (as seen in Figure 7(b)) the objective of maximizing local accuracy is aligned with reducing the local fairness gap. Therefore, collaboration improves local fairness, with personalization further enhancing the model's local accuracy and therefore its fairness.

We also note that (local) accuracy and fairness may not necessarily be aligned. Our next experiment shows that personalization can still improve fairness in such tasks compared to non-personalized `FedAvg`, which (we interpret) is driven by a combination of statistical and computational benefits. Specifically, we conduct experiments on another task, ACSIncome, with "sex" as the protected attribute. Fig 8(a) shows that for this

task, the fraction of samples is comparable across groups for label 0 data, but differs for label 1 data. From Fig 8(c), we observe that this time, all collaborative training algorithms improve accuracy but have *worse* local fairness compared to `Standalone` learning; this is because improving (local) accuracy is not aligned with fairness in this task. That said, we observe that the personalized FL algorithms slightly improve local fairness compared to `FedAvg`. We interpret this as the statistical advantage of (effectively) observing more label 1 data (as `FedAvg` does, too), combined with a computational advantage of not overfitting a global model to the majority label 0 data (unlike what `FedAvg` may be doing).

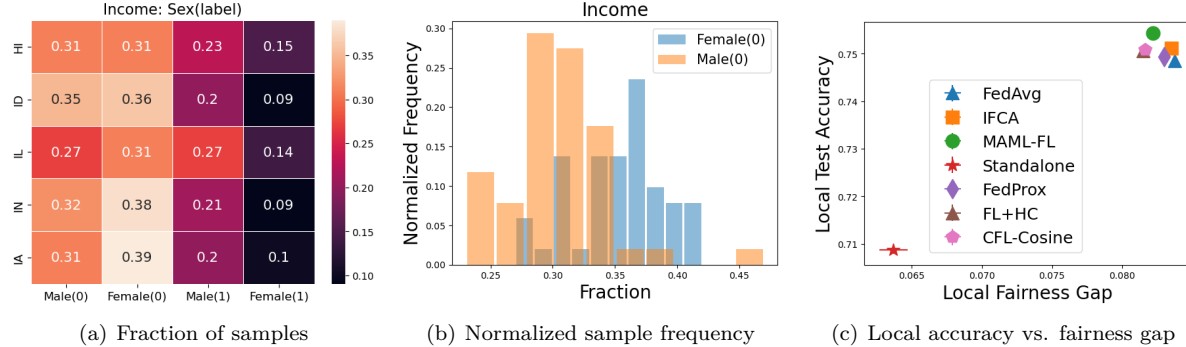

| (a) Fraction of samples | (b) Normalized sample frequency | (c) Local accuracy vs. fairness gap |

Figure 8: Experiments on the ACSIncome task with sex as the protected attribute.

### 4.3 Experiments on the `Adult` dataset

We also contrast these methods on the `Adult` dataset (Dua & Graff, 2017). Among those 48842 samples, 41762 samples belong to the White group, while 7080 samples are from the Non-White groups. Given the nature of this data's heterogeneity, we randomly generate 5 clients each with an unbalanced number of samples based on race. Additionally, in Appendix E.5, we conduct experiments where samples are distributed with less heterogeneity across 5 clients, as done in other existing FL studies (e.g. (Ezzeldin et al., 2023)).

| Client ID | Non-White samples | White samples |
|-----------|-------------------|---------------|
| 0 | 300 | 1000 |
| 1 | 300 | 1000 |
| 2 | 300 | 39362 |
| 3 | 600 | 200 |
| 4 | 5580 | 200 |

| (a) Number of samples | (b) Local accuracy vs. fairness gap |

Figure 9: Experiments on the `Adult` dataset with race as the protected attribute.

From Fig 9(b), we can see that the results are consistent with our findings in Section 4.1. Interestingly, we observe that the `IFCA` algorithm clusters clients by grouping those with more White samples into one cluster and those with more Non-White samples into another (i.e., {0,1,2}, {3,4}). In contrast, the `FL+HC` clusters clients by grouping those with more samples into one cluster and those with less samples into another (i.e., {0,1,3,4}, {2}). As a result, these two variants of clustering-based algorithms have different performance, having statistical advantages but for different reasons.

### 4.4 Analytical support

To further support our numerical findings, we analytically show that personalized federated clustering algorithms, which group similar clients together to enhance local accuracy, can improve local fairness compared to a non-personalized global model under certain conditions. Specifically, we consider a setting with

single-dimensional features that lead to optimal threshold-based classifiers and assume clients are grouped into two clusters $\{C_\alpha, C_\beta\}$ based on similarities in their local datasets.

Let $\theta_G^*$ and $\theta_i^*, i \in \{\alpha, \beta\}$ denote the optimal decision threshold for the FedAvg algorithm and for clusters $C_\alpha$ and $C_\beta$, respectively. We define $\Delta_f(\theta_C^*)$ as the $f$-fairness performance, $f \in \{\texttt{EqOp}, \texttt{SP}, \texttt{EO}\}$ when using personalized federated clustering algorithms. Specifically, it quantifies the overall $f$-fairness achieved by applying the optimal decision threshold $\theta_i^*$ to its corresponding cluster $C_i$. Similarly, we define $\Delta_f^i(\theta_i^*)$ as the $f$-fairness measured within cluster $C_i$.

**Proposition 1** (Improved overall `EqOp` through clustering). *Let $\mathcal{G}_g^{y,c}(x)$ be unimodal distributions for $y \in \{0,1\}, g \in \{a,b\}, c \in \{C_\alpha, C_\beta\}$, with modes $m_g^{y,c}$ satisfying $m_b^{y,c} \leq m_a^{y,c}, \forall g, c$, and $\alpha_b^{1,C_\alpha} \geq \alpha_g^{0,C_\alpha}, \forall g$. Suppose $\theta_\alpha^* < \theta_\beta^*$. Then, there exists a cluster size $\hat{p}$ such that for $p \geq \hat{p}$, we have $\Delta_{EqOp}(\theta_C^*) \leq \Delta_{EqOp}(\theta_G^*)$,*

*where $\hat{p}$ is the solution to $\hat{p} = \min\{1, |\frac{\int_{\theta_G^*}^{\theta_\beta^*} \mathcal{G}_a^{1,\beta}(x) - \mathcal{G}_b^{1,\beta}(x) dx}{\int_{\theta_\alpha^*}^{\theta_G^*} \mathcal{G}_a^{1,\alpha}(x) - \mathcal{G}_b^{1,\alpha}(x) dx}|\}$ and $\theta_G^*$ is obtained as: $\theta_G^* = \arg\min_\theta \hat{p} *$*

*$\sum_{j \in \mathcal{C}_\alpha} \mathcal{L}_j(\theta) + (1 - \hat{p}) * \sum_{j \in \mathcal{C}_\beta} \mathcal{L}_j(\theta)$.*

The proof is provided in Appendix D.1. Intuitively, clients in $C_\alpha$ benefit from their personalized model because increasing the decision threshold (i.e., shifting from $\theta_\alpha^*$ to $\theta_G^*$) reduces the true positive rate of the disadvantaged group $b$ faster than that of the advantaged group $a$, increasing the fairness gap in $C_\alpha$. For clients in $C_\beta$, the opposite effect occurs. However, under the given conditions, the fairness improvement in $C_\beta$ is insufficient to compensate for the fairness degradation in $C_\alpha$, resulting in a unfairer outcome when using $\theta_G^*$.

**Proposition 2** (Improved `SP` for $C_\alpha$ through clustering). *Let $\mathcal{G}_g^{y,c}(x)$ be Gaussian distributions with equal variance for $y \in \{0,1\}, g \in \{a,b\}, c = C_\alpha$, with means $\mu_g^{y,c}$ satisfying $\mu_g^{0,c} \leq \mu_g^{1,c}, \forall g$. Suppose $\theta_\alpha^* < \theta_\beta^*$. If $\alpha_g^{1,c} \geq \alpha_g^{0,c}, \forall g$ and $\alpha_a^{0,c} \exp(\frac{(\bar{\theta} - \mu_a^{0,c})^2}{-2\sigma^2})(\bar{\theta} - \mu_a^{0,c}) - \alpha_b^{1,c} \exp(\frac{(\bar{\theta} - \mu_b^{1,c})^2}{-2\sigma^2})(\bar{\theta} - \mu_b^{1,c}) \geq \alpha_b^{0,c} \exp(\frac{(\bar{\theta} - \mu_b^{0,c})^2}{-2\sigma^2})(\bar{\theta} - \mu_b^{0,c}) - \alpha_a^{1,c} \exp(\frac{(\bar{\theta} - \mu_a^{1,c})^2}{-2\sigma^2})(\bar{\theta} - \mu_a^{1,c})$, where $\bar{\theta} := \frac{\mu_a^{1,c} + \mu_b^{0,c} + \mu_b^{1,c} + \mu_a^{0,c}}{4}$, then there exists a $\hat{p}$ such that for $p \geq \hat{p}$, $\Delta_{SP}^\alpha(\theta_\alpha^*) \leq \Delta_{SP}^\alpha(\theta_G^*)$.*

The proof is provided in Appendix D.2, and we can reach similar conditions for the cluster $C_\beta$. Intuitively, when there are more label 1 samples in both groups, the global model $\theta_G^*$ will pull the $C_\alpha$ cluster model $\theta_\alpha^*$ up to account for the label imbalance, resulting in a deterioration in both fairness and accuracy for clients in this cluster. Note that Proposition 2 considers the `SP` fairness, which is impacted by both the group $a$ vs. $b$ feature distributions *as well as* the label rates, rendering it more stringent than `EqOp` fairness of Proposition 1. Figure 10 illustrates this by

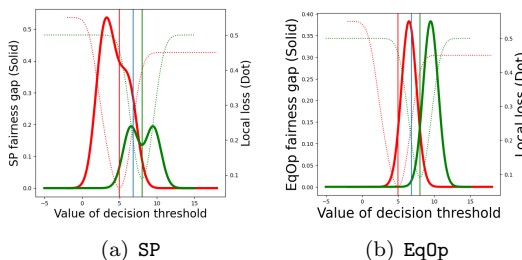

(a) `SP`      (b) `EqOp`

Figure 10: Fairness gap vs $\theta$.

plotting the fairness gap vs. the decision threshold $\theta$ for `SP` vs. `EqOp`, showing that `SP` exhibits less structured changes as the decision threshold moves (e.g., due to the use of a global model).

## 5 Conclusion

We proposed new fairness-aware federated clustering algorithms, `Fair-FCA` and `Fair-FL+HC`, which take both fairness and accuracy into account when clustering clients. Our methods effectively span the space between personalized FL and locally fair FL. We find that they can lead to improved local fairness (matching or exceeding existing locally fair FL methods, without any explicit fairness interventions) while allowing for a tunable trade-off between accuracy and fairness at the client level. We have also both numerically and analytically shown that there can be (unintended) fairness benefits to personalization in FL, which our clustering-based fair FL algorithms are exploiting whenever possible. Identifying methods to integrate fairness considerations into other (non-clustering based) personalized FL algorithms, and extending our analytical findings (both for clustered FL algorithms, and to other classes of personalized FL methods), are main directions of future work.

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
