# OpenReview forum: "Friends in Unexpected Places: Enhancing Local Fairness in Federated Learning through Clustering"
_TMLR — Withdrawn by Authors_

### Review · Reviewer_3QrB · 2025-12-16

**Summary Of Contributions:**

The authors proposed Fair-FCA, which extends the existing personalized learning framework IFCA by incorporating a fairness formulation.  The authors also proposed Fair-FL + HC that is based on a pretrained FedAvg framework. Fairness is evaluated using three different metrics.

The proposed method is suitable for scenarios in which the degree of heterogeneity across clients is high, such that learning local models for individual clients (or small groups of clients) is more beneficial than training a single global model. It is also applicable when sensitive attributes are present in the clients’ data distributions.

Strengths:

Overall, the method section is well written and easy to follow. The proposed method is interesting and intuitive. I also appreciate the investigation of the method from multiple perspectives ($\gamma=0$, $\gamma \in (0, 1)$, and $\gamma=1$)

Weaknesses:
* The two local fairness baselines are unpublished methods, which makes it difficult to assess the contribution of the paper in terms of improving local fairness. This is particularly problematic because improving local fairness is presented as the main contribution of the work.
* The significance of the experimental results in Figures 2 and 3 is difficult to judge without confidence interval in both dimensions. In several cases, the performance differences appear to be very small (e.g., on the order of 0.01%), making it hard to determine whether the authors’ claims are supported by the results. Please provide consistent significance measures for both accuracy and fairness gap across all experiments.
* It is somewhat confusing that the results on synthetic data are presented in the main paper, while the results on real-world datasets are placed in the appendix, especially for section 3.4. When looking at the results on the real world dataset in section E.3 in appendix, the performance improvement is very limited, e.g., sometimes only 0.1% performance improvement.
* Some experimental results are difficult to interpret. For example, the fairness gap of FedAvg is slightly lower than that of the personalized method IFCA (which is a special case of the proposed approach). This suggests that personalization does not necessarily improve fairness. Similarly, the results shown in Figures 6, 7, and 8 are not entirely consistent, indicating that personalization may improve fairness only in certain cases.
* The problem is formulated as a binary classification task, and the sensitive attribute used for fairness evaluation is also binary. It is unclear whether the proposed method can scale to multi-class classification problems and fairness evaluations based on multi-class sensitive attributes.
* In Figure 1, I can understand that locally fair federated learning may achieve better local fairness at the expense of lower local accuracy compared to personalized federated learning. However, it is unclear why personalized federated learning would necessarily exhibit worse fairness than locally fair federated learning. This figure appears to be motivated primarily by Figure 5, which shows the results based on synthetic data. Therefore, it is unclear whether this trend would hold for real-world datasets or in more general settings.
* How is k chosen in practice?
* The choice of setting $\gamma = 0$ is unclear. When $\gamma = 0$, a client is assigned the model that minimizes the fairness gap. It is not clear why this choice is optimal, especially given that the remainder of the algorithm does not explicitly involve fairness optimization.


Minor comments:
* Please standardize the number of decimal places on the y-axis across all figures and ensure that the figures are presented with consistent sizing (e.g., Figure 3).

**Audience:**

Yes

**Audience Explanation:**

Improving fairness with respect to sensitive attributes is important, so I believe that at least some individuals will be interested in this paper

**Claims And Evidence:**

No

**Claims Explanation:**

The authors state that clustering-based methods can improve local fairness. Therefore, the authors should compare their approach with federated learning methods that are explicitly proposed to improve local fairness. However, the two local fairness methods used for comparison are both unpublished and focus on both local and global fairness.

The performance improvement is sometimes on the order of 0.01% or 0.1%, which appears to be limited.

Regarding the claim that personalization can improve fairness, Figure 6 shows that FedAvg even achieves a smaller fairness gap than the method specifically designed for personalization, which suggests that personalization does not necessarily or consistently lead to fairness improvements.

**Requested Changes:**

See the first four items in the weakness section.

---

### Review · Reviewer_Gvbt · 2026-01-27

**Summary Of Contributions:**

The paper suggests ways to improve fairness guarantees in a federated setting by moving beyond central fairness measures. Specifically, simple global objectives achieve uniform fairness, but in a heterogeneous federated setting, these objectives can collapse for clients with features considerably outside the average. Thus, the authors propose improving local fairness by changing which clients collaborate with whom, rather than adding explicit fairness constraints during training or post-processing predictions. Furthermore, to inculcate the idea of local fairness, the authors in Algorithm 1 update the objective to model a weighted system considering both the usual Iterative Federated Clustering Algorithm (IFCA) and a fully local federated learning system controlled by $\gamma$.

The second algorithm leverages a hierarchical clustering-based solution that relies on the same weighted global objective, reducing intra-cluster variance and then running FedAvg. Empirically, the authors claim that clustering clients while accounting for local fairness can match or exceed existing locally fair FL methods such as EquiFL and PPFL, while avoiding explicit fairness interventions. They also show that personalization alone (i.e., letting $\gamma = 1$) can improve local fairness in some regimes.

### Main Contributions:
Fairness-aware cluster assignment for clustered/personalized FL, with a single knob $\gamma$ that spans personalized and locally fair behavior. Two separate algorithms to achieve the same objective.
Empirical suggestions that personalization can have “unintended fairness benefits”, supported by analytical evidence for simplified settings.

**Audience:**

Yes

**Audience Explanation:**

The paper tackles the idea of delivering fairness in a federated setting, a claim that directly impacts collaborative training, privacy implications, and thus is important for the TMLR audience.

**Claims And Evidence:**

Yes

**Claims Explanation:**

The claims are primarily backed by two separate evidence streams:

1. The benefits of Fair-FCA and Fair-FL+HC are supported with experiments comparing the current algorithm's performance with existing methods in Figures 2, 3, and 5.
2. For the personalization claim, the analytical details in section 4.4 alongside the experiments showcased in Figures 7, 8 support the idea that personalized, collaborative learning improves a client's fairness factor by making all clients better.

**Requested Changes:**

Below, the authors can find possible ways to enhance their results and thus provide further compelling evidence for their methods.

Cluster stability / Assignment dynamics analysis

- Report how often clients switch clusters across rounds
- Showcase stability vs $\gamma$, vs fairness metric and vs K.
- Include some more details about the switch rate over each round, including what these training dynamics mean

Dynamics for when clients have extreme data distributions

- What happens when denominators are small (e.g., few positives in a group) for fairness measures like EO/EqOp?
- Ablation showing performance as client dataset size shrinks (to see how fairness gap changes with client data volumes)

For Fair-FL+HC, how is the pairwise fairness matrix computed efficiently? Is it approximated?

Privacy Leakage:
- What is transmitted to the server (beyond what FedAvg already shares)?
- Can cluster membership leak sensitive-attribute prevalence or label rates? Possible solutions?

Generalization
- How can the method scale to multi-class labels?
- How many minimum clients are required for the fairness mechanisms to function properly?
- How does cluster hyperparameter tuning occur, and what is the overhead for the same?

---

### Review · Reviewer_YoZs · 2026-02-06

**Summary Of Contributions:**

This paper considers a federated learning setting in which each client performs a binary classification task under fairness constraints (Statistical Parity, Equal Opportunity, and Equalized Odds). The authors propose algorithms designed to balance local accuracy and local fairness. The approach builds on existing clustering-based personalized federated learning methods by incorporating a fairness metric into the cluster assignment procedure.

More precisely, the clustering is based on the following client-specific loss:
$\gamma L(Z_i,\theta_k) + (1-\gamma) \Psi(Z_i,\theta_k)$,
where $\gamma \in [0,1]$ controls the trade-off between accuracy and fairness, $Z_i$ denotes the data of client $i$.

The main contribution of the proposed method is to account for heterogeneity across clients and to enable information sharing among clients with similar data distributions while explicitly controlling fairness. The parameter $\gamma$ allows one to interpolate between purely accuracy-driven clustering and fairness-aware clustering.

The authors propose two algorithms and illustrate their performance on both synthetic and real datasets. They also consider the case $\gamma = 1$ , which recovers existing methods such as IFCA and FL+HC. The empirical results suggest that clustering-based approaches can reduce unfairness.

**Additional Comments:**

I am also interested in a variant where clients are first clustered with $\gamma = 1$ (purely accuracy-based) and fairness is enforced or approximated in a second step. This seems related to the discussion in Section 3.2.2. Could the authors comment on this approach and its potential advantages or limitations?

**Audience:**

Yes

**Audience Explanation:**

The paper focuses on the crucial problem of balancing accuracy and fairness in the federated learning framework. Therefore, it fits well within the scope of TMLR and should be of interest to the machine learning community.

**Broader Impact Concerns:**

Since the paper deals with fairness, I think that a broader impact statement should be included in the manuscript.

**Claims And Evidence:**

No

**Claims Explanation:**

While the proposed method is interesting and shows promising performance, I have several concerns, particularly regarding the clarity of the paper and positioning of the paper. See my comments below


1)  In Section 3.1 (Problem Statement), it would be helpful to further explain the objective of federated learning and why clustering-based personalization is a relevant strategy to achieve this objective.

2) The numerical experiments are sometimes difficult to follow. For example, in the first experiments, the distribution of the six clients and the number of clusters used are not clearly specified.

3) The evaluation protocol is not sufficiently detailed. Information about training/test splits, number of repetitions, and evaluation metrics should be clearly stated.

4) For the Adult dataset, the sample size per client is not reported. Such details should be included in the main body of the paper.

5) While the first algorithm is clearly presented, the description of the second algorithm is somewhat vague. For instance, the sentence “and performs several local updates to personalize their model” lacks precision and would benefit from further elaboration.

6) The paper sometimes reads more like an illustration of how to add a parameter $\gamma$  to existing methods than a fundamentally new methodological contribution. The authors could strengthen their contribution by more clearly articulating the novelty and theoretical implications of their approach.

**Requested Changes:**

While the paper provides interesting methods to trade-off  local accuracy and local fairness in the federated learning setup, the paper suffers from its lack of clarity (see my comments above) and also a discussion on the choice of parameter $K$ (the number of clusters) should be conducted (see my comments below).

1) Regarding the number of clusters $K$: it is unclear whether
 is assumed to be known or treated as a hyperparameter. If it is a hyperparameter, the authors should study the impact of misspecifying
$K$ and possibly propose a method for selecting it.
2) Could the authors comment on the sensitivity of the proposed methods to the number of clients and clusters? The experiments seem to focus on moderate-scale settings, and it would be useful to understand how the approach scales.
3) Propositions 1 and 2 are interesting but are restricted to the univariate case. Could these results be extended to the multivariate setting, which is more relevant in practice?

---

### Comment · Action_Editor_4L3Q · 2026-02-19
**Reviews are out**

Dear Authors,

We have now received three reviews of your submission. Please take the time to read them carefully and provide responses to the reviewers’ comments and concerns as appropriate.

If you have any questions or specific concerns regarding your submission, please do not hesitate to reach out.

All the best,
The AE

---

### Note · Authors · 2026-02-19

**Comment:**

We thank the reviewers, and the AE, for their time and feedback.

**Withdrawal Confirmation:**

I have read and agree with the venue's withdrawal policy on behalf of myself and my co-authors.